# An Overview on Anodes for Magnesium Batteries: Challenges towards a Promising Storage Solution for Renewables

**DOI:** 10.3390/nano11030810

**Published:** 2021-03-22

**Authors:** Federico Bella, Stefano De Luca, Lucia Fagiolari, Daniele Versaci, Julia Amici, Carlotta Francia, Silvia Bodoardo

**Affiliations:** Department of Applied Science and Technology, Politecnico di Torino, Corso Duca degli Abruzzi 24, 10129 Turin, Italy; s269980@studenti.polito.it (S.D.L.); lucia.fagiolari@polito.it (L.F.); daniele.versaci@polito.it (D.V.); julia.amici@polito.it (J.A.); carlotta.francia@polito.it (C.F.); silvia.bodoardo@polito.it (S.B.)

**Keywords:** magnesium battery, anode, Sn-Bi alloy, post-Li battery, Mg metal

## Abstract

Magnesium-based batteries represent one of the successfully emerging electrochemical energy storage chemistries, mainly due to the high theoretical volumetric capacity of metallic magnesium (i.e., 3833 mAh cm^−3^ vs. 2046 mAh cm^−3^ for lithium), its low reduction potential (−2.37 V vs. SHE), abundance in the Earth’s crust (10^4^ times higher than that of lithium) and dendrite-free behaviour when used as an anode during cycling. However, Mg deposition and dissolution processes in polar organic electrolytes lead to the formation of a passivation film bearing an insulating effect towards Mg^2+^ ions. Several strategies to overcome this drawback have been recently proposed, keeping as a main goal that of reducing the formation of such passivation layers and improving the magnesium-related kinetics. This manuscript offers a literature analysis on this topic, starting with a rapid overview on magnesium batteries as a feasible strategy for storing electricity coming from renewables, and then addressing the most relevant outcomes in the field of anodic materials (i.e., metallic magnesium, bismuth-, titanium- and tin-based electrodes, biphasic alloys, nanostructured metal oxides, boron clusters, graphene-based electrodes, etc.).

## 1. Introduction

The high concentration of CO_2_ in the atmosphere is causing a temperature rise never recorded before, due to its strong greenhouse effect [1,2,3]. This problem arises from decades of increased use of fossil fuels for energy purposes, with tons of CO_2_ consequentially released into the atmosphere, formed by carbon atoms which were previously stocked underground [4,5,6]. Even today, most of the energy is generated from fossil sources [7,8,9]. Furthermore, due to the demographic and/or economic growth of some areas of the world, it is expected that by 50 years the energy consumption is destined to double [10]. In a similar framework, action must be taken immediately and effectively to prevent an uncontrolled increase of the average temperature of the planet. Fossil fuels must be replaced by clean energy sources, which does not involve greenhouse gas emissions, and a significant amount of attention is paid towards renewable energy solutions. However, renewable energies have a major disadvantage, which greatly limits their diffusion, i.e., their poor predictability [11,12]. It may, therefore, happen that, when the energy demand is high, production through renewables is weak, or that there is a surplus of production at times with low demand. Moreover, it is possible that consumption by users of an electricity grid based on photovoltaic energy mainly occurs after sunset [13,14]. 

In this scenario, California could represent a case study, being a state historically launched towards a wide use of renewable energy sources, photovoltaics in particular [15,16,17,18,19]; the above-mentioned issues were highlighted some years ago [20,21,22]. In the chart shown in Figure 1A, it is possible to appreciate the difference between the demand for electricity and the electric power produced by photovoltaic panels in the Californian grid, in a typical spring day and depending on the various years from 2012 to 2020 [23,24]. An approximatively constant delta is observed in the early hours of the morning, which then decreases as sun rises and photovoltaic production begins. This tendency is increasingly emphasized as years pass, until the spread undergoes a real collapse moving towards 2020. In the central hours of the day, there is even an over-generation risk, as photovoltaic power rapidly increases and sums to the other means of production already in action, with the consequence that the instantaneously generated energy might be higher than the demand, which is highly dangerous for the electricity grid. In the final hours of the day, the tendency is reversed: in a very short time, the demand for electricity goes up, while photovoltaic panels stop producing, so that just where a peak of the demand is observed it is no longer possible to rely on solar energy. Due to the shape composed by the curves, which resemble a duck, this graph has been called the “duck curve” [25,26], and highlights a problem that actually goes beyond the borders of California.

Similar problems are common to other kinds of renewable energies [30,31,32,33,34]. It is, therefore, clear that, in order to reach a correct integration of renewables, a way to store energy when in excess and to use it when needed is necessary [35,36,37,38,39]. Still referring to the “duck curve”, a significant step forward in terms of efficiency and reduction of CO_2_ emissions would be made if the excess of photovoltaic electricity produced in the middle of the day was stored and used after the sunset, concurring to satisfy the high demand of the evening and avoiding the over-generation risks [40,41,42,43,44]. This is what is highlighted in the graph shown in Figure 1B.

Rechargeable batteries, being devices capable of reversibly converting electricity into chemical energy, stand among the most suitable technologies to accomplish this task [45,46,47,48,49]. Their use is not limited to the purposes highlighted above, but is destined to become increasingly massive also in the automobile industry [50,51,52,53,54]. The number of hybrid or electric cars produced is getting bigger and bigger [55,56,57]. In Figure 1C, the evolution of the global car fleet is quantified, according to three different scenarios. BAU stands for "business as usual" and represents an extension of the current trend over time. “BAU-F2” and “BAU-F4” represent scenarios in which the CO_2_ emissions of the transport sector are divided by 2 and by 4 compared to 2005, respectively. It is, therefore, clear that the role of transport electrification will be fundamental in order to reduce the emissions [29].

To date, a large number of rechargeable batteries exists, bearing different characteristics, uses, advantages and disadvantages. Some of the most important are classified in Table 1 [10].

However, all the batteries listed above are far from allowing the existence of a society based on renewable sources, where a large amount of energy is stored for residential and transport sectors. Depending on the type, there are disadvantages linked to efficiency, cost, toxicity or safety, which severely limit a mass use as needed [10].

Among all types of batteries, lithium-ion batteries (LIBs) play a crucial role in the evolution of modern technologies [61,62,63,64,65]: they are used in laptops, cell phones, electric vehicles and many other devices [66,67,68,69,70]. Made with graphitized carbon as anode material and a transition metal oxide as cathode, they are able to accumulate 240 Wh kg^−1^ or 640 Wh L^−1^ for thousands of cycles [71]. They will probably drive technological progress for many other years, as no better batteries (in terms of energy density and lightweight) will be available in the near future. The LIB, however, possesses multiple drawbacks [72,73,74,75,76]. One of the main limitations is the relatively scarce concentration of lithium in the Earth’s crust [77,78], that in addition is mostly located in a few countries (Bolivia and Chile owe more than 50% of global resources, as depicted in Figure 2). Difficulties with current technology in disposing the exhausted LIBs and in recovering lithium from them at reasonable prices make the situation worse [79,80,81]. Production cost, despite all the progresses made, is still quite high, and also represents an obstacle. Finally, there are problems related to safety of use of the battery: over time, the anode degrades giving rise to dendritic formations that may lead to short circuits, overheating and possible battery explosion [10,82,83,84,85].

Given all the reasons mentioned above, scientists are looking for new types of batteries (the so-called “new chemistries” [87,88,89,90,91]). One of the most interesting solutions seems to be represented by the rechargeable magnesium-ion batteries (MIBs) [92,93,94,95,96], which utilize magnesium cations as the active charge transporting species in solution and (in many cases) metallic magnesium as the anode. A primary advantage of this technology is given by the solid magnesium anode that leads to high energy density values, well above those of lithium-based cells [97,98,99,100,101]. However, some issues have emerged when using elemental magnesium and novel solutions have been proposed. In this mini review, we will highlight the current pros and cons of MIBs, with a special focus on the role of metallic magnesium anodes and the most reliable alternatives when the upscaling of this technology (e.g., for large-scale energy storage coupled with renewables) is conceived. In addition, the nanodimensionality of the proposed anodic materials and its effect on the electrochemical behaviour of the resulting MIBs will be highlighted, by discussing case studies based on nanotubes, nanoparticles, nanopores, nanocrystals, nanoflakes and nanowires.

## 2. Rechargeable Magnesium-Ion Batteries: State of Art

With reference to the scheme shown in Figure 3, the MIB device is not different with respect to the corresponding lithium or sodium counterparts. Magnesium metal has huge potentialities to serve as an anode material for rechargeable batteries, starting from its theoretical volumetric capacity of 3832 mAh cm^−3^, clearly superior to that of metallic lithium (2061 mAh cm^−3^) [102]. Moreover, although lithium has a higher mass capacity, the chemistry of magnesium does not lead to dendrite formation, considerably improving the safety of devices where these batteries are used [103]. The greatest advantage of magnesium lies on its abundance, being one of the most abundant elements in the Earth’s crust. This would benefit production costs and availability of the supply, because the extraction of magnesium cannot risk being monopolized by a small number of countries as in the case of lithium, and because it is possible to rely on larger amount of raw material. Table 2 shows the average abundance, expressed in ppm, of some elements in the Earth’s crust; the comparison between lithium and magnesium is noteworthy [104].

Moreover, another advantage, considering a future battery industry and the entire supply chain, of magnesium versus lithium is its perfect recyclability. In fact, with respect to other metals, magnesium can be recycled without any degradation of its physical properties. The energy cost for recycling and melting processes is also lower than that required for recycling other metals and is approximately 5% of the cost required for the production of crude [107,108], and better performances of recycling are achieved. This is also clear from Figure 4, that shows the end-of-life recycling rate of some elements of the periodic table, i.e., the ratio between the amount of element truly recycled and the total quantity of element introduced in the recycling flow (noteworthy comparing the 25–50% of magnesium vs. less than 1% for lithium).

Despite these advantages, turning MIBs into a marketable solution is not trivial at all, due to several technical challenges. The biggest problem is the strong tendency of magnesium to passivate in a wide variety of solvents, salts and contaminants, so that any kind of electrochemical reaction is blocked and magnesium deposition/dissolution process is not reversible. Another major challenge consists in coupling Mg with a high-voltage/high-capacity cathode in which the anode can behave reversibly. Many cathodic materials, capable to reversibly store lithium ions, do not work with magnesium ions, mainly because of their high charge density, caused by the divalent character of the ions, with a small radius, leading to strong and detrimental interactions with the host material. These two problems are enough to dramatically reduce the number of available electrolytes and cathodes, making the research extremely difficult [110].

According to the current state-of-the-art MIB components [105,111,112,113,114], the anode may be realized using magnesium metal or some alternative materials. Among those, some of the most studied and promising are bismuth and tin. As for the cathode, four of the most important families have been identified: cobalt, vanadium, molybdenum and manganese-based cathodes [110,115,116,117,118]. It is of primary importance, for a good cathode, to reversibly host magnesium ions and allow their high mobility within the electrode matrix, assuring the anode compatibility with the electrolyte. Solutions based on Grignard reagents, organoborate, borohydride and Mg(TFSI)_2_ are some of the most studied electrolytes [119,120,121,122]. Important parameters that guide the electrolyte selection are the resistance to oxidation, the Coulombic efficiency (i.e., a measure of the reversibility of charge deposition), fast charge transport, the behaviour in the presence of contaminants and the volatility. Innovative solutions such as solid magnesium electrolytes have also been proposed, aiming at overcoming the issue of volatility, while keeping possible a good charge transport [71].

## 3. Magnesium Metal as Anode

As said, magnesium possesses very interesting properties. On one hand, it is theoretically capable of storing up to 3832 mAh cm^−3^ of charge [102], and its high reactivity imparts to the metal with the desired virtue of a significantly negative voltage. Even though the nature of the passivating layer has not been fully understood, it is known that its formation comes from the high reactivity of magnesium metal, which acts as a double-edged sword: electrolytes are instable in proximity of the anode and their decomposition occurs [123]. The passivating nature of this layer contrasts with what is observed when analogous electrolytes are in contact with lithium metal [124,125,126,127,128]. In this case, the formation of the layer, the solid electrolyte interphase (SEI), allows the diffusion of lithium ions and prevents further decomposition of the electrolyte in the highly reducing environment during lithium plating [129,130,131,132,133].

The research on suitable electrolytes is not a simple task, and hampers the overall development of the battery [134,135,136,137,138]. As depicted in Figure 5, electrolytes based on magnesium salts, for example perchlorates and tetrafluroborates, and polar aprotic solvents, like carbonates and nitriles, form a passivating layer [103]. This explains why the choice of the electrolyte is limited to a few possibilities. An example is given by Grignard-based electrolyte solutions, that, however, suffer of limited anodic stability [139,140].

Anyway, an important advantage of magnesium anode is related to the morphology of magnesium deposits. Unlike lithium [141,142,143,144,145], plating magnesium from organohalo-aluminate electrolyte does not involve dendrites formation. Moreover, the morphology of magnesium deposits from magnesium aluminate complexes has been related to the current densities applied during deposition. No dendritic formations have been observed, but, as shown in Figure 6, there is a preferred orientation of the deposits, depending on the current density. For instance, (001) orientation is the preferred one at low current densities, while the (100) is preferred at high current densities. Consequently, it is possible to hypothesize that both thermodynamic stability and diffusion rates of Mg ions govern the crystals growth of magnesium depositions [71,146].

It is expected that advancements in the understanding of magnesium chemistry will lead to the achievement of advanced SEI layers, similar to those formed in traditional LIBs [147,148,149,150,151]. To date, it is of crucial importance that the anode remains free from any solid passivation layer in order to allow reversible stripping/plating. Thus, only chemically stable electrolytes are viable in batteries that use such anode materials. Furthermore, a selection has to be made, as many compatible electrolytes cannot be used in practical systems due to issues of safety, low electrochemical stability window and high cost.

Compatibility among magnesium anode, electrolyte and cathode should also be taken into account. At first, a large part of the electrolyte solutions that allowed reversible magnesium deposition had a very low anodic stability window, with a maximum of 2–2.4 V vs. Mg^2+^/Mg. This narrow voltage limitation is not acceptable with high-voltage metal oxide cathodes. Recently, electrolyte formulation without Grignard reagents, which are able to reach the anodic stability up to 3.3 V, have been proposed. Nonetheless, most of them contain chlorides, which may be corrosive to metal oxide cathode materials and current collectors [110]. In addition, 1,2-dimethoxyethane (DME)—an important solvent in non-aqueous electrolytic solutions—hinders intercalation of magnesium ions into V_2_O_5_ [152] cathode material.

## 4. Strategies beyond Elemental Magnesium Anodes

The use of alternative anodic materials has been recently proposed, so that conventional electrolytes may be employed [153,154,155,156,157]. These are magnesium ion insertion anodes, composed by an alloy of magnesium combined with other metals. The principle behind this choice is that magnesium alloys, if thermodynamically favourable, should lower the reductive power of the anode. As a consequence, such materials will have less negative potential, that may lead to anodes chemically consistent with electrolytes and some contaminants too. As a result, a wide variety of electrolytes would be available for reversible magnesium alloying-dealloying reactions. To this purpose, new materials should satisfy some requirements, that can be summarized as follows:Cheap, ubiquitous, eco-friendly and safe materials;Highly reversible alloying–dealloying process;Sufficiently fast magnesium diffusion, or phase propagation, within the base metal;High energy density;The voltage difference between alloying-dealloying processes must be as small as possible;Compatibility with inert components, such as conducting additives, binders and current collectors must be assured.

Despite these drawbacks, the possible electrode pulverization, as consequence of volume changes, and sluggish magnesium insertion/extraction kinetics, insertion anodes are gaining interest as alternatives to magnesium metal [71,110].

### 4.1. Bismuth-Based Anodes

Bismuth-based anodes are among the most studied alternative anodes, due to the rhombohedral crystalline structure of bismuth that allows formation of alloys with high volumetric capacity [158]. The gravimetric capacity theoretically achievable is also high, about 385 mAh g^−1^, hypothesizing the transfer of six electrons and the formation of the alloy according to the following reaction [159]:2Bi + 3 Mg^2+^ + 6 e^−^ → Mg_3_Bi_2_(1)

An interesting superionic conductivity of magnesium ions in β-Mg_3_Bi_2_ has been shown [160]. The electrochemical behaviour of this alloy and some of its derivatives was studied by Matsui et al. [158], by adopting bismuth, antimony and Bi_1−x_Sb_X_ alloys at different stoichiometries as anode materials for MIBs. Antimony showed very poor cycling performances, but bismuth and Bi_0.88_Sb_0.12_ alloy exhibited impressive results at a current density corresponding to 1C (see Table 3). The capacity fading detected by the authors was probably due to losses of electrical contact caused by periodic volume change of the anode during cycling.

Shao et al. studied the electrochemical characteristics of bismuth nanotubes (NTs), synthesized by hydrothermal reaction [161]. The purpose was to reduce the diffusion length of magnesium ions in order to mitigate the kinetic hurdles that exist in alternative anode materials. The experiments were conducted for both bismuth NTs and microparticles, in Mg(BH_4_)_2_ + LiBH_4_ diglyme electrolytes. Figure 7 displays the cyclic voltammograms (CV), the discharge/charge profile and the rate/cycling performances of the cells assembled by Shao et al. The results clearly showed superior performances in the case of a bismuth nanostructured anode, especially at high C rates (Figure 7c). At 5C, the specific capacity was 216 mAh g^−1^ for bismuth NTs vs. 51 mAh g^−1^ for bismuth microparticles. The former also showed narrower peaks and lower overpotential (Figure 7A), together with a faster response. At low C rates, i.e., between C/20 and C/2, the performances of the two structures were almost comparable. Furthermore, bismuth NTs were studied in a full cell setup, by choosing Mo_6_Se_8_ as cathode and an electrolyte consisting of Mg(TFSI)_2_ 0.4 M in diglyme. A mid-point discharge voltage of about 0.75 V and a specific capacity of 90 mAh g^−1^ were observed. The full cell also exhibited a 92.3% capacity retention after 200 cycles (Figure 7D). X-ray diffraction (XRD) analysis confirmed the high reversibility of magnesium ions insertion. A weakness, though, was the large voltage hysteresis between magnesium alloying and dealloying, which resulted in energy losses. Other drawbacks were the exotic nature of the anodic material and its manufacturing costs, which made it difficult to be used in large scale commercial systems [110].

The bismuth-based anode was also studied by Murgia et al. by electrochemical measurements coupled with XRD [162]. The experiments were conducted in two-electrode cells, which contained metallic magnesium as counter and reference electrodes and an organometallic-based electrolyte solution. An unexpected phenomenon was observed: a biphasic process occurred between bismuth and Mg_3_Bi_2_ without any intermediate amorphization, that is the rule for alloy-type electrodes. Micrometric bismuth and Mg_3_Bi_2_ prepared by ball-milling delivered specific capacity of 300 mAh g^−1^ at a discharge rate of 2C, with Coulombic efficiency of 98.5% after 50 cycles (Figure 8A). Moreover, a full cell composed by a Mg_3_Bi_2_ anode and a Mo_6_S_8_ cathode in a conventional electrolyte solution of Mg(TFSI)_2_ 0.5 M in dyglime was developed. The full cell showed a voltage profile with a discharge plateau at around 0.6 V. Both the intercalation process on the cathodic side and the de-alloying process of the anode during discharge were corroborated through ex situ XRD measurements. However, full de-magnesiation of Mg_3_Bi_2_ was not achieved.

Moreover, by ^25^Mg nuclear magnetic resonance spectroscopy, aimed at understanding the mechanism and diffusion pathway for magnesium ions in the bismuth anode [163], two-phase alloying reactions of magnesium and bismuth were demonstrated, and such spectroscopy studies enlightened a fast exchange between the two magnesium sites in the Mg_3_Bi_2_ alloy.

Di Leo et al. proposed the synthesis of bismuth/carbon nanotubes (CNTs) composite [159]. Electrochemical deposition of bismuth on CNTs from aqueous solution of Bi(NO_3_)_3_ was adopted to obtain the composite material. They observed a specific capacity of 180 mAh g^−1^ through CV at a rate of 0.5 mV s^−1^ in acetonitrile-based solution containing Mg(ClO_4_)_2_ 0.5 M and dipropylene glycol dimethyl ether 0.5 M. However, the capacity decreased to 80 mAh g^−1^ at the second cycle and to 49 mAh g^−1^ at the third. This sharp capacity fading excluded the material from any further investigation.

Some computational studies confirmed the great potential of bismuth-based materials to serve as anodes, despite some limits. A computational study by Jin et al. underlined that the diffusion barrier for an isolated magnesium ion in bismuth was 0.67 eV, a relatively low value, which does not change with the alloying level. This suggests the existence of a concrete possibility to obtain a fast charging/discharging magnesium battery, based on magnesium-bismuth alloys as anode [164].

### 4.2. Tin-Based Anodes

Another promising family of materials for MIBs alternative anodes is that of tin-based electrodes. Tin theoretically implies several advantages over bismuth as anode material [110]: (1) lower insertion potential; (2) higher theoretical specific capacity; (3) considerably lower atomic weight (118.71 u for tin vs. 208.98 u for bismuth); (4) Earth abundancy of tin is approximatively two orders of magnitude greater than that of bismuth. Furthermore, in the case of bismuth-based anodes each bismuth atom is able to exchange three electrons (see Equation (1)).

As for tin-based anode, the anodic alloy is Mg_2_Sn and its chemical reaction allows an exchange of four electrons for each tin atom, one more than what occurs for each bismuth atom:Sn + 2 Mg^2+^ + 4 e^−^ → Mg_2_Sn(2)

Finally, tin offers better performances in terms of recycling than bismuth, as shown in Figure 4 (tin > 50%, bismuth < 1%).

One of the first studies on this topic was that of Singh et al., based on tin powder films for insertion anodes [165]. By plotting galvanostatic charge–discharge curves for both tin- and bismuth-based anodes at a low current density of C/500 in an organo-haloaluminate electrolyte, the superior electrochemical performances of tin were put in light. The insertion potential into tin anode was found to be equal to 0.15 V vs. Mg^2+^/Mg, against 0.23 V vs. Mg^2+^/Mg for the bismuth anode, and a hysteresis between insertion and de-insertion of 50 mV was observed for tin, much lower than that of 90 mV for bismuth. The tin anode showed an initial impressive specific capacity of 903 mAh g^−1^, but, unfortunately, the discharge process revealed to be highly irreversible, with a sharp reduction in reversible capacity (Figure 8B). Rate capability measurements also showed that, at rates above C/500, the specific capacity of the tin anode rapidly decreased. This was possibly due to a poor insertion kinetics of the tin anode, even though the Coulombic efficiency seemed to increase with the charge–discharge rate. The sharp reduction of initial capacity was also observed in full cell setups, with Mg(TFSI)_2_ 0.5 M in DME/organohaloaluminate electrolyte and Mo_6_S_8_ cathode. The resulting electrochemical performances were quite similar for both systems: 82 mAh g^−1^ at the first cycle, followed by a stable value of less than 50 mAh g^−1^ for the following ones [165].

Beyond pure tin, some tin-based compounds were also studied. Cheng et al., for example, focused on a tin–antimony alloy by means of a combined computational and experimental approach [166]. In their first study, they discovered that, during the first cycle, an irreversible process led to the formation of a porous structure composed of antimony-rich and pure tin sub-structures. After this initial phase (conditioning), they observed that the nanosized tin particles had a highly reversible behaviour, while the antimony-rich zones showed low Coulombic efficiency due to trapping. Thus, the antimony-rich zones lowered the specific capacity of the anode, but they seemed to be necessary to reach formation of stable tin nanoparticles [166]. Another study focused on the behaviour of the tin–antimony alloy after conditioning led to the conclusion that the alloy had superior properties than that of pure tin [167]. Overall, the advantages of the tin–antimony alloy can be summarized as follows: (1) improved kinetics for magnesiation/demagnesiation result in lower overpotentials (Figure 9A); (2) improved specific capacity at the same current density (420 mAh g^−^^1^ vs. less than 300 mAh g^−1^ for pure tin at a 50 mA g^−1^) (Figure 9B); (3) excellent rate capability with 70% capacity retention (300 mA g^−1^ at very high current densities of 1000 mA g^−1^ (Figure 9C); (4) good cyclability, with 270 mAh g^−1^ after 200 cycles at a current density of 500 mA g^−1^ (Figure 9D).

Wang et al. used density functional theory (DFT) calculations to study magnesium cation diffusion properties in β- and α-Sn. They found a diffusion barrier for an isolated magnesium atom of 0.395 eV in the α-Sn and of 0.435 eV in the β-Sn. Moreover, a higher magnesium concentration decreased the diffusion barrier in the case of α-Sn, while an opposite behaviour was expected for β-Sn. Thus, the α form of tin seemed to represent a better alternative than the β phase as an anode material for MIBs [168].

### 4.3. Biphasic Bismuth–Tin Alloy Anodes

As said above, both bismuth and tin anodes show interesting properties, but also have some drawbacks. Bismuth anode does not deliver highly specific capacities, while the tin anode experiences strong capacity fading and irreversible chemistry, due to the sluggish solid-state diffusion of magnesium ions and slow charge transfer at the interface [169]. Interfacial design would be a good way to improve ionic transport properties, as phase/grain boundaries act as channels for magnesium ions. Designing biphasic or multiphase alloys, rather than single phase, would be a clever strategy to increase the density of phase/grain boundaries, so that a better transport of magnesium ions would be allowed and kinetics of the electrode would be improved [170].

Biphasic bismuth–tin alloys were proposed and discussed in a recent study [170], expected to combine the high capacity of tin with the good reversibility of bismuth. A facile dealloying strategy was developed, with an alternate phase distribution and a nanoporous structure (NP) thanks to which volume expansion is mitigated and diffusion length is shortened. During the discharge process, bismuth and tin consecutively react with magnesium ions to form Mg_3_Bi_2_ and Mg_2_Sn, respectively, while the charge process brings to de-magnesiation of Mg_3_Bi_2_ and Mg_2_Sn to regenerate bismuth and tin. High performances were observed, mainly thanks to the porous structure, the dual-phase microstructure and the high density of phase/grain boundaries. Figure 10A schematically shows the increased grain boundaries and magnesium ions transport channels on the atomic scale that is possible to obtain after the first cycle in a dual-phase bismuth–tin electrode. Two different alloys were tested, i.e., Bi_6_Sn_4_ and Bi_4_Sn_6_, synthesized by chemical dealloying of rapidly solidified Mg_90_Bi_6_Sn_4_ (at %) and Mg_90_Bi_4_Sn_6_ (at %) precursor ribbons in a 2 wt % tartaric acid solution at ambient temperature. Scanning electron microscopy (SEM) images of the two alloys are shown in Figure 10B,C. Their theoretical specific capacities were 623 and 525 mAh g^−1^, respectively.

MIB performances with these alloys were studied in an all-phenyl-complex 0.4 M electrolyte. Cells were composed of a magnesium foil and the alloy as electrodes, and comparison with NP-bismuth and NP-tin electrodes was also carried out. Figure 10D shows the resulting CV traces, at the scan rate of 0.05 mV s^−1^ for NP-bismuth and alloy electrodes, and at 0.01 mV s^−1^ for NP-tin within the voltage range of 0–0.6 V vs. Mg^2+^/Mg. It is possible to observe a larger peak area for the NP-Bi_4_Sn_6_ electrode than that for the NP-Bi_6_Sn_4_ one, suggesting a higher specific capacity. For the two alloys, a two-step reversible magnesiation and de-magnesiation reaction occurred, associated to two couples of redox peaks. This was attributed to the biphasic nature of the alloys. The relatively small redox peaks of the NP-tin electrode were explained by its low reactivity and inferior kinetics. Galvanostatic discharge/charge profiles of the NP-bismuth and alloy electrodes (at a current density of 50 mA g^−1^) and of NP-tin sample (at 20 mA g^−1^) within the voltage range of 0–0.8 V vs. Mg^2+^/Mg (Figure 10E) showed that NP-Bi_6_Sn_4_ and NP-Bi_4_Sn_6_ electrodes managed to deliver high values of discharge specific capacities equal to 434 and 482 mAh g^−1^, respectively. These values were much higher than those of NP-bismuth and NP-tin, i.e., 330 mAh g^−1^ at 50 mA g^−1^ for NP-bismuth and only 31 mAh g^−1^ even at 20 mA g^−1^ for NP-tin. These results were consistent with the redox peaks displayed by the CV.

Galvanostatic profiles also helped to understand the chemical processes that occurred during discharge and charge of the biphasic bismuth–tin electrode. During discharge, it was possible to observe two plateaus at around 0.16 and 0.23 V vs. Mg^2+^/Mg. They were associated to the magnesiation processes successively occurring in bismuth and tin. During charge, a sloped plateau (0.25–0.32 V vs. Mg^2+^/Mg) followed by a second one at about 0.33 V vs. Mg^2+^/Mg were the expression of the de-magnesiation processes and the regeneration of tin and bismuth. Biphasic electrodes were characterized by a much lower hysteresis between magnesiation and de-magnesiation curves if compared to both NP-bismuth and NP-tin samples, due to decreased polarization effect. Finally, it was possible to extrapolate the contribution of bismuth and tin to the final specific capacity of NP-Bi_4_Sn_6_ and NP-Bi_6_Sn_4_ (see Table 4). The alloy with the highest content of bismuth was expected to achieve superior performances because of the relatively small volume changes of bismuth and faster diffusion kinetics, with formation of the superionic conductor Mg_3_Bi_2_.

Rate capability was also analysed (Figure 11A). The results showed excellent rate capability performances for NP-Bi_6_Sn_4_. In addition, its discharge capacity of 434 mAh g^−1^ at 50 mA g^−1^ gradually decreased with increasing current densities, but at 1000 mA g^–1^ it was still equal to 362 mAh g^−1^ (Figure 11B). The tin phase was responsible of the majority of capacity fading with increasing currents. At 1000 mA g^−1^, the bismuth phase capacity contribution showed a 6% fading with respect to its contribution at 50 mAh g^−1^, against 30.3% for tin. A similar trend was observed for NP-Bi_4_Sn_6_. Figure 11C also displays that the Coulombic efficiency of NP-Bi_6_Sn_4_, initially relatively low (94.5%), gradually increased as the rate raised: the reason was attributed to the reduction of reacting quantity of tin, which was responsible of the capacity fading. At 1000 mA g^−1^, because of the higher content of tin, NP-Bi_4_Sn_6_ showed a specific capacity of 260 mAh g^−1^, but it must be considered that capacity of NP-bismuth collapsed to 7 mAh g^−1^ under the same current density. The tin phase was thus essential to reach good performances at high current densities. As concerns cycling stability, the bismuth–tin alloy showed excellent results. Figure 11C displayed, for comparison, cycling performance of NP-bismuth and alloy samples at 200 mA g^–1^, while that of NP-tin was recorded at 20 mA g^−1^. Figure 11D shows discharge/charge profiles of NP-Bi_6_Sn_4_ for different cycles at 200 mA g^−1^.

Bismuth–tin alloy stability was clearly superior. NP-Bi_6_Sn_4_ specific capacity decreased from 412 mAh g^−1^ at the first cycle to 280 mAh g^−1^ at the 200th, namely 68% capacity retention. Thanks to the discharge/charge profiles (Figure 11D), it was possible to extrapolate the role that bismuth and tin played in the capacity fading, as shown in Table 5. The results interestingly showed that, upon cycling, the specific capacity of the bismuth phase decayed faster than that of tin phase: this was probably caused by the dual-phase nature of the electrode, which unlocked the kinetics limitations of the tin phase, leading to a highly reversible chemistry. As for NP-Bi_4_Sn_6_ electrode, specific capacity rapidly decreased from 374 to 336 mAh g^−1^ during the initial 10 cycles, resulting in 10% capacity fading. This sharp difference with the behaviour of NP-Bi_6_Sn_4_ was attributable to the different amount of bismuth and tin in the alloy; indeed, a part of tin failed to participate in the reaction after 10 cycles, at 200 mAh g^−1^. It was suggested that NP-Bi_4_Sn_6_, in which the tin phase was dominant, could hardly accommodate large volume changes with high current densities during the initial cycles, as the volume change of 214% during the phase transformation between tin and Mg_2_Sn was larger than that of 100% between bismuth and Mg_3_Bi_2_. After 200 cycles, NP-Bi_4_Sn_6_ capacity decreased from 374 to 220 mAh g^−1^, with 58.8% capacity retention (vs. 68% for NP-Bi_6_Sn_4_). However, the Coulombic efficiency of both the biphasic alloys was very high, approximatively 99% after 30 cycles, because of the highly reversible behaviour of the two phases of bismuth and tin (Figure 11C).

The chemical processes occurring within the electrode during charge and discharge are sketched in Figure 11E, and can be summarized as follows. During discharge (alloying), at first magnesium ions reacted with bismuth to form Mg_3_Bi_2_, while the unreacted tin behaved as a buffer matrix mitigating volume expansion. Then, magnesium ions reacted with tin and the previously formed Mg_3_Bi_2_ played the role of preventing further volume expansion. The dual phase unlocked the potential properties of tin, that otherwise would be characterized by poor kinetics and low electrochemical reactivity. Meanwhile, Mg_3_Bi_2_ acted as a superionic conductor [160], accelerating the magnesium ions transport and explaining the better rate performances of NP-Bi_6_Sn_4_, in which the bismuth phase was dominant. During the first charge (dealloying) process, magnesium ions were extracted from the electrode, bringing to formation of smaller-sized bismuth and tin nanocrystals. While the process occurred, Mg_3_Bi_2_ and tin served as a buffer matrix to prevent large volume shrinkage. Formation of the buffer matrix during charge and discharge, the unlocked potential of tin and the behaviour of Mg_3_Bi_2_ as a superionic conductor, combined with the effects of the porous structure and the increased density of grain/phase boundaries, explained all the good properties provided by the dual-phase bismuth–tin alloys [170].

### 4.4. Titanium-Based Anodes

Titanium-based anodic materials have been extensively studied for lithium- and sodium-based batteries [171,172,173,174,175], together with a broad examination of nanostructures and morphologies able to guarantee mechanical stability, high performance, ease of preparation on metal supports and efficient charge transport in one-dimensional materials. However, studies on titanium-based anodes in the field of MIBs are still few and the scientific community started publishing the first results in 2020.

Luo et al. proposed layered sodium trititanate (Na_2_Ti_3_O_7_) and sodium hexatitanate (Na_2_Ti_6_O_13_) nanowires (NWs) as anodes for MIBs [176]; they were prepared by heat treatment of the titanate precursor under different washing conditions. The investigation highlighted that passing from the layered Na_2_Ti_3_O_7_ morphology to a more condensate three-dimensional microporous structure in Na_2_Ti_6_O_13_ boosted the magnesium ions storage performance. The two electrodes exhibited a NWs-based morphology (Figure 12A,B); transmission electron microscopy (TEM) images showed that Na_2_Ti_3_O_7_ displayed a large interlayer spacing of 0.84 nm, while for Na_2_Ti_6_O_13_ was equal to 0.75 nm. Overall, the interlayer distance was rather large, enough to guarantee an efficient magnesium ions storage and diffusion. Na_2_Ti_6_O_13_-based cells achieved initial discharge and charge capacity of 165.8 and 147.7 mAh g^−1^, respectively, at 10 mA g^−1^, with an outstanding initial coulombic efficiency of 89.1% (Figure 12C). The electrochemical reaction mechanism was investigated through different techniques, from which it emerged that the inserted magnesium ions replaced the sites of sodium ions to form Mg–Ti–O. In this system, sodium ions could not reinsert into the structure because of the formation of insoluble NaCl particles. Such an irreversible structure change and NaCl salt formation led to a rapid worsening of Na_2_Ti_3_O_7_-based cells specific capacity values. On the other hand, Na_2_Ti_6_O_13_, with its regular three dimensional and microporous structure based on TiO_6_ octahedra, guaranteed better structural stability during the magnesium ions insertion and extraction processes.

Yang et al. fabricated an original magnesium-ion dual-ion battery adopting expanded graphite as cathode and Ti-doped Nb_2_O_5_ nanoflakes (Ti–Nb_2_O_5_ NFs) as anode [177]. The latter was designed with the aim of shortening the diffusion distance of Mg^2+^ and consisted of a hierarchical structure including microspheres (diameter: 4–5 μm) assembled by nanoflakes (Figure 13A–C). As a further innovative point, ether solvents with high inflammability and narrow electrochemical window were excluded from the electrolyte, and replaced with an ionic liquid-based solution containing Mg(TFSI)_2_ 0.5 M in Pyr_14_TFSI (with additives). The resulting dual-ion battery (i.e., during charging magnesium ions intercalated into the Ti–Nb_2_O_5_ NFs, while TFSI^–^ ions intercalated into an expanded graphite, see Figure 13D) showed a high discharge capacity (93 mAh g^−1^) at 1C and a capacity retention of 79% after 500 cycles at 3C (Figure 13E) and 77% at 5C. Such a very good rate performance coupled with a rather high discharge medium voltage of ≈ 1.83 V; the latter resulted in a high energy density (174 Wh kg^−1^ at 183 W kg^−1^ and 122 Wh kg^−1^ at 845 W kg^−1^), currently the best values in the field and that could be successfully led to device integration with solar cells [178,179,180,181,182].

### 4.5. Other Materials

In this section, we list other metals (and related compounds) used in the last two years as anodes for MIBs. It is important to underline that also computational chemistry aims to predict new promising compounds as anodes for MIBs; some of these examples are discussed below.

Zhang et al. prepared VO_2_ NWs by a conventional hydrothermal process [183]; their width was uniformly distributed around 100–300 nm and the length was equal to ≈ 10 μm (Figure 14B). This electrode was first tested in a MgSO_4_ 1.0 M aqueous electrolyte, showing initial charged capacities equal to 263, 207.7, 146.4 and 103 mAh g^−1^ at 100, 200, 500 and 1000 mA g^−1^, respectively. Conversely, performances in MgCl_2_- and Mg(NO_3_)_2_-based electrolytes were rather poor, thus showing—again in the MIBs field—the relevant role of the magnesium salt for electrolyte formulation. As concerns prolonged charge/discharge experiments at 500 mA g^−1^, the VO_2_ NWs revealed a cycling stability of 54.3% after 100 cycles (Figure 14A); such an important fading was attributed to the partially irreversible reaction of magnesium ions (not being fully extracted from the host material lattice) and the partial dissolution of VO_2_ in the electrolyte upon time. Even improvements were required, the authors were able to follow the mechanism of the reaction by different characterization techniques. In detail, VO_2_ transformed into a stable MgVO_x_ structure after the first charge, then the insertion/extraction of magnesium ions was accompanied by the valence changes of V^5+^ (reduced to V^4+^) and V^4+^ (reduced to V^3+^).

Shakerzadeh et al. went through the latest outcomes in the field of boron clusters, among which the recently experimentally observed all-boron B_40_ fullerene with D*_2d_* symmetry is a stable allotrope. The authors set up DFT calculations to determine if B_40_ fullerene could behave as an efficient anode for MIBs [184]. It emerged that the interactions between magnesium ions and heptagonal/hexagonal holes of B_40_ fullerene were much stronger than those with magnesium metal. To calculate the performance of B_40_ fullerene, its six holes were decorated with magnesium metal centres (Figure 14C), resulting in an open-circuit voltage and a storage capacity of 5.5 V and 744 mAh g^−1^, respectively. Surprisingly, it also emerged that halides (fluoride, chloride, bromide) encapsulation within B_40_ fullerene could markedly enhance the open-circuit voltage up to 8.8 V. Overall, this study should push experimental scientists towards the preparation of boron-based anode materials for MIBs.

Theoretical investigations were also carried out to predict the fabrication of MIBs in stretchable configurations, in order to be employed in healthcare devices and sensors. In particular, Wu et al. worked on two-dimensional nitrogenated holey graphene (C_2_N) anodes (Figure 14D) [185], predicting a maximum capacity and an open circuit voltage of 1175 mAh g^−1^ and 0.447 eV, respectively, in the strain-free state. The authors highlighted that the mechanical activation represents an effective strategy to boost charge redistribution and raising capacity values under tensile and compressive strains. Overall, it emerged that strains were able to remodulate—at the atomic scale—the structure of anodic materials, redistributing electrons in a uniform way and promoting the reactivation of adsorption sites. The stretchable device showed a two-stage diffusion mechanism: the out-of-plane magnesium ions diffused rapidly in the first stage, while in the following one the in-plane magnesium ions migrated moderately. Even if this study seems quite far from the traditional ones in the field of MIBs, it was precious to provide new insights on microscale mechanisms for stretchable energy storage devices, suggesting suitable structural requirements for bidimensional anodes.

Several others anodic materials have been recently proposed and require further validation within the scientific community. Just to mention some of them, indium and lead were studied. After some experiments, indium has been judged inadequate due to high costs and low rate capability [186]; furthermore, an electrochemically driven amorphization of crystalline MgIn takes place when In is combined with Pb in solid solution [187]. Lead was excluded, even before considering the associated environmental issues and toxicity, because of the poor Coulombic efficiency [188]. The path ahead towards stable, efficient and cheap anodes for MIBs is still long, but the overall scenario towards large-scale energy systems able to store electricity from solar energy [189,190,191,192] is becoming feasible.

## 5. Conclusions

Despite the impressive properties shown by magnesium metal, its utilization as anode material has been shown to be unpractical mainly because of its very high sensitivity to surface reactions, which makes the choice of the electrolyte extremely difficult.

It is a matter of fact that, compared to the magnesium metal anode, all the ion insertion anodes cannot compete in terms of specific capacity; however, because of the problems caused by magnesium metal, their utilization seems to be necessary to conceive a commercial device. The main benefit brought by ion insertion anodes is the possibility to use conventional and well-known electrolytes that support reversible magnesium deposition. The cathode-side should also be taken into account, as it must be capable to reversibly work with the electrolyte solution. 

Alternative anodes, allowing the utilization of a wider range of suitable electrolytes, ensure a better compatibility with cathodes. Moreover, as cathodes often represent the limiting factor in batteries, moving from magnesium metal anode to alternative anodic materials, with the associated loss of anodic specific capacity, may still result advantageous in terms of energy density of the whole device if oxide cathodes are used. Some materials, such as bismuth and tin, exhibited good properties like low reduction potentials and relatively high theoretical specific capacities, but currently they are far from being used in practical systems because of their poor electrochemical stability during prolonged cycling. Thus, dual-phase alloy anodes have been proposed by several groups as a possible solution and, in many cases, they showed superior properties if compared with both bismuth and tin separately, mainly thanks to better kinetics and the high reversibility that the double phase provides. Although the investigated dual-phase bismuth–tin alloys cannot be considered as a definitive solution, they anyway indicate an important field of research that may lead to the development of an advanced electrode material in the future.

Future perspectives in this field could pass through two materials classes that are leading to noteworthy results in the energy storage field. First, functionalized MXenes/graphene heterostructures could exhibit a great potential for magnesium ions intercalation, keeping at the same time a low interlayer expansion (and an overall reduction of geometric constraint). A second idea could be that of designing carbon-based anodes, following some strategies already developed for aqueous batteries; for example, carbon molecular sieve constructed by filling a carbon source into a sacrificial template could make use of the mesoporous tunnels of the template as a frame to host more magnesium ions.

Other important research fields to promote the development of MIBs (also targeting higher TRL levels) concern the preparation of chlorides-free electrolyte solutions with wide electrochemical windows, where magnesium can behave reversibly, and the elaboration of new cathodes, less sensitive to the composition of the electrolyte solutions. There is still much work to do towards the utilization of a clean and more efficient battery in order to support the ongoing worldwide energy transition.

## Figures and Tables

**Figure 1 nanomaterials-11-00810-f001:**
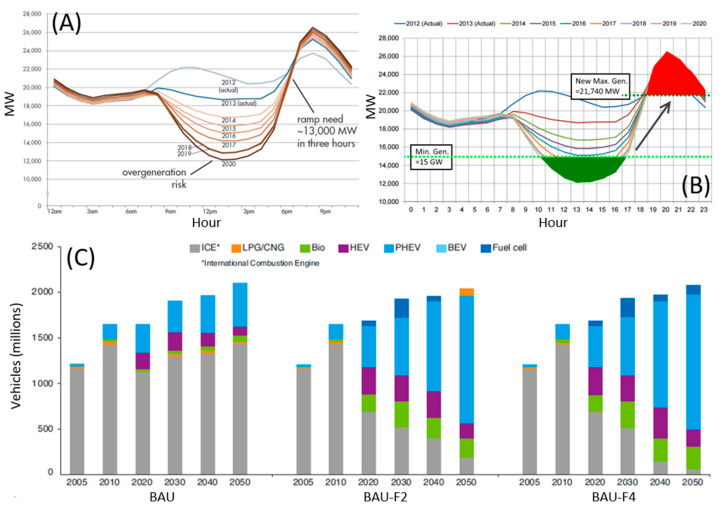
(**A**) Difference between electricity demand and photovoltaic electricity produced during a typical spring day and in different years in the Californian grid; (**B**) “Duck curve” showing energy storage to capture energy during overgeneration risk period (green area) and discharging said energy during peak net load hours (red area); (**C**) Evolution scenarios of the global car fleet, including combustion vehicles (ICE), liquefied or compressed natural gas (LPG/CNG), hybrid (HEV), plug-in hybrid (PHEV), electric with batteries (BEV), electric with fuel cells (Fuel cell). Adapted with permission from [27,28,29]. Copyright California ISO (2013), Stanford University (2015) and IFP Energies Nouvelles (2018).

**Figure 2 nanomaterials-11-00810-f002:**
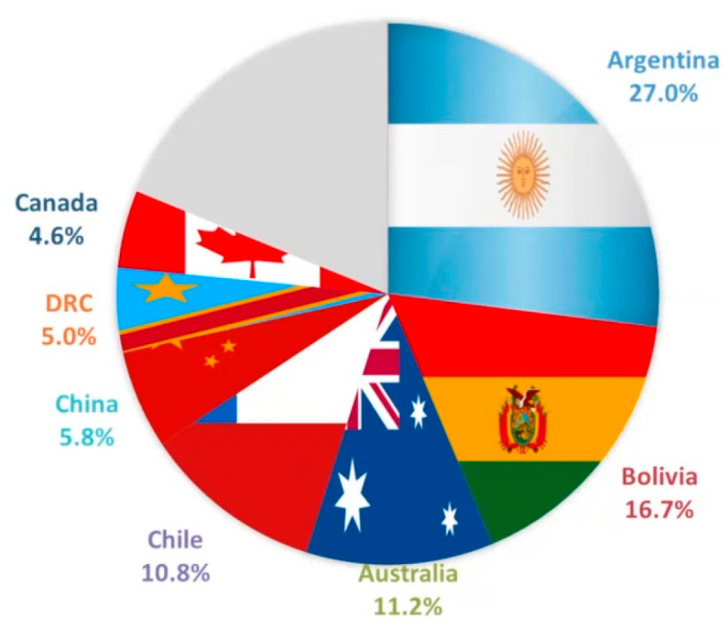
Distribution of lithium resources in 2019. Adapted with permission from [86]. Copyright Edison, 2019.

**Figure 3 nanomaterials-11-00810-f003:**
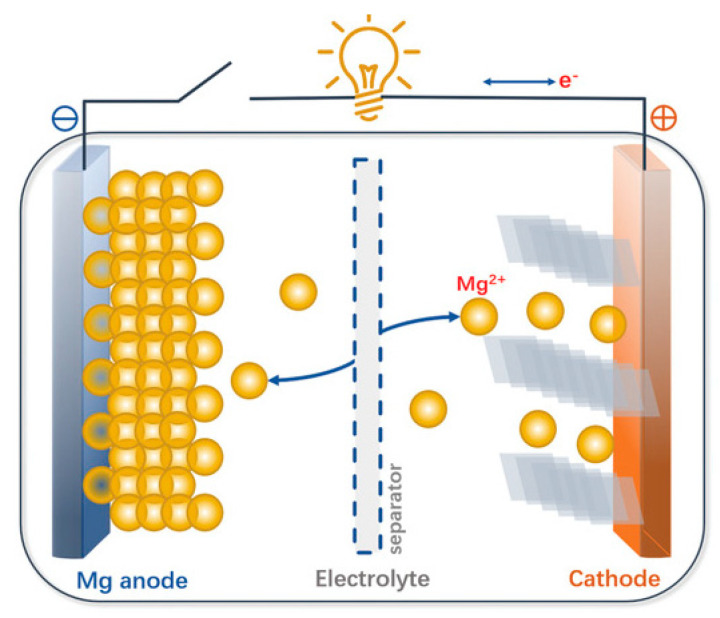
Scheme and working principle of a magnesium rechargeable battery. Adapted with permission from [105]. Copyright John Wiley & Sons, Inc., 2020.

**Figure 4 nanomaterials-11-00810-f004:**
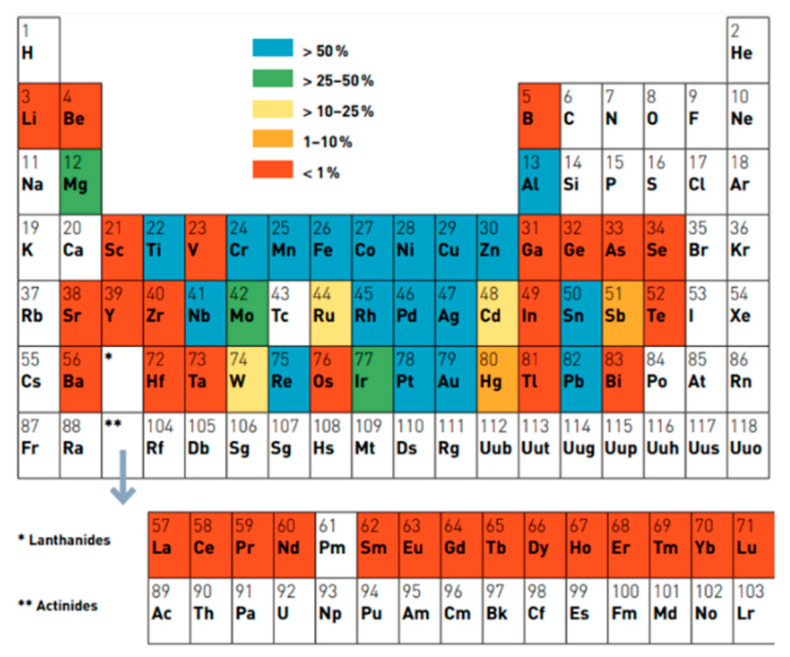
End-of-life recycling rate of some elements of the periodic table. Adapted with permission from [109]. Copyright UNEP, 2011.

**Figure 5 nanomaterials-11-00810-f005:**
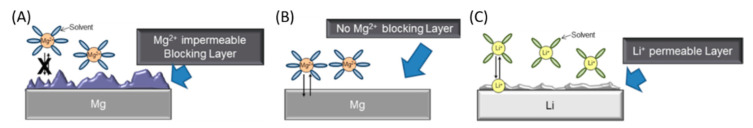
The metal/electrolyte interfaces for (**A**,**B**) magnesium- and (**C**) lithium-based systems. Different from lithium, magnesium experiences passivation when the metal is exposed to conventional electrolytes (case A, e.g., with ionic salts and polar solvents), while magnesium passivation does not occur in ethereal organo-magnesium electrolytes (case B, e.g., with Grignard-based solutions). Adapted with permission from [71]. Copyright Beilstein-Institut, 2014.

**Figure 6 nanomaterials-11-00810-f006:**
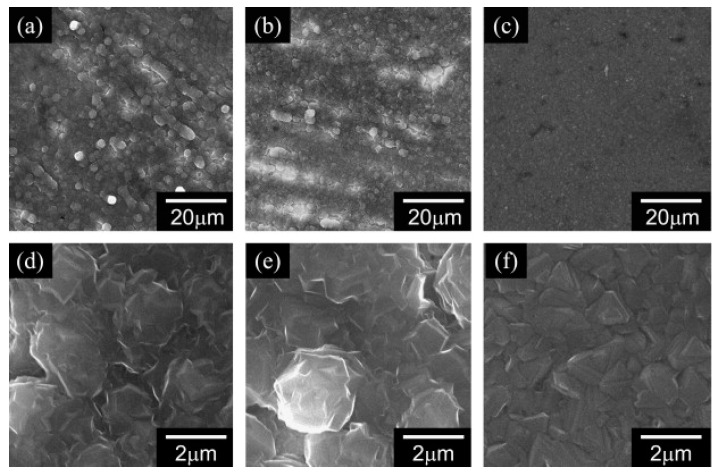
Scanning electron microscopy (SEM) micrographs of electrodeposited magnesium at different current densities: (**a**) 0.5 mA cm^−2^; (**b**) 1.0 mA cm^−2^; (**c**) 2.0 mA cm^−2^; (**d**) 0.5 mA cm^−2^; (**e**) 1.0 mA cm^−2^ and (**f**) 2.0 mA cm^−2^. Adapted with permission from [146]. Copyright Elsevier B.V., 2011.

**Figure 7 nanomaterials-11-00810-f007:**
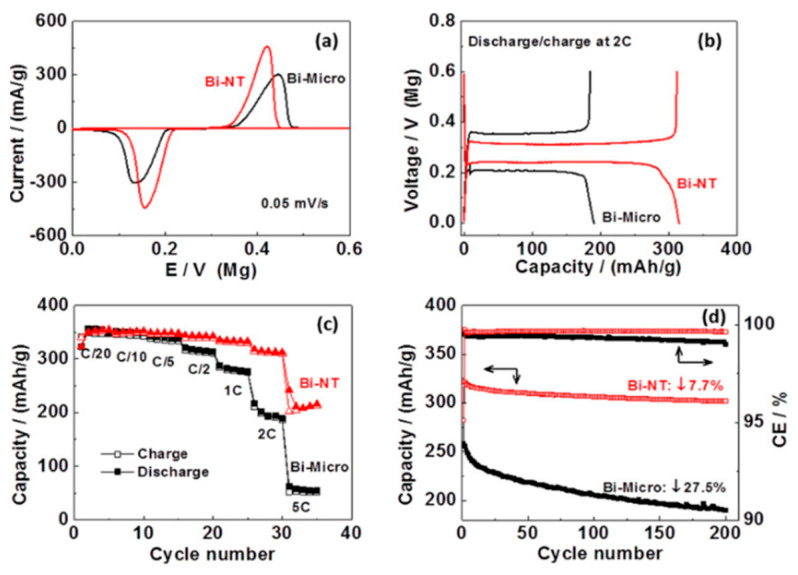
Electrochemical performances of bismuth NTs as anodes for MIBs: (**a**) Cyclic voltammograms (CV) of magnesium ions insertion/deinsertion; (**b**) Discharge/charge profile of a cell; (c) Rate performance of a cell; (**d**) Cycling stability and Coulombic efficiency of bismuth NTs for reversible magnesium ions insertion/deinsertion (C-rate was not reported by the authors). Cell configuration: Mg/Mg(BH_4_)_2_ 0.1 M + LiBH_4_ 1.5 M in diglyme/Bi. A comparison with the corresponding microstructured anodes is also shown in each plot. Adapted with permission from [161]. Copyright American Chemical Society, 2014.

**Figure 8 nanomaterials-11-00810-f008:**
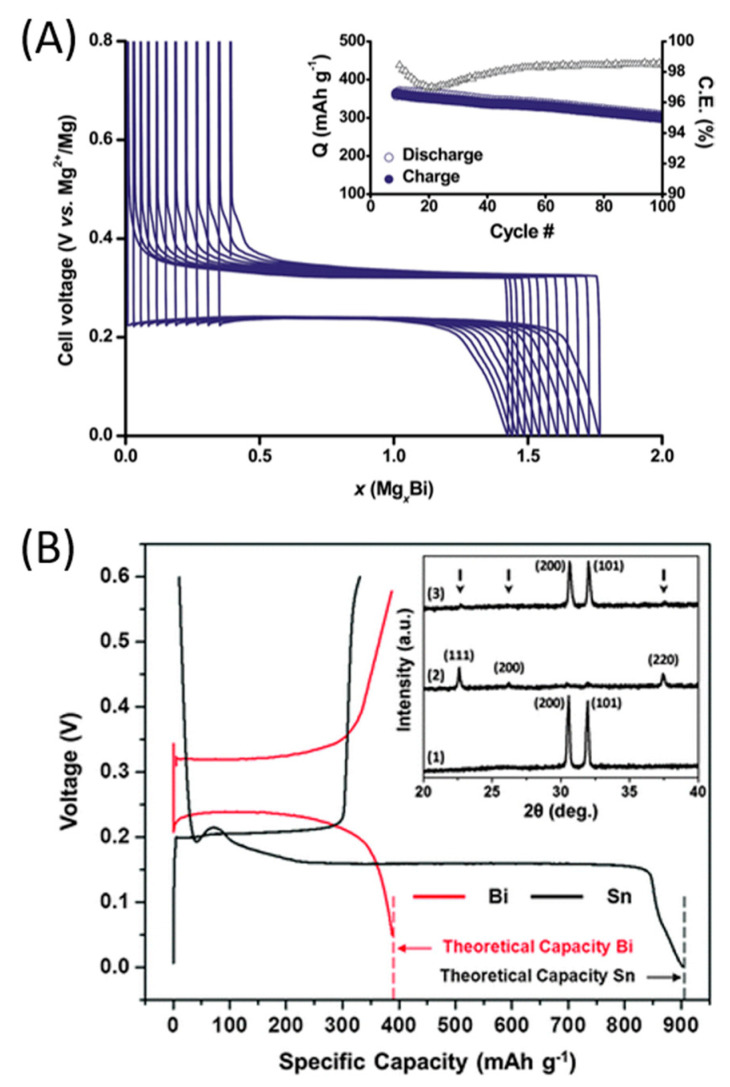
(**A**) Galvanostatic curve at 2C, after initial activation sweeps, obtained with copper foil supported electrode based on micrometric bismuth particles embedded by carbon additives. Inset: evolution of discharge and charge capacities and Coulombic efficiency. (**B**) First cycle galvanostatic magnesiation/de-magnesiation curves for Sn/Mg and Bi/Mg half cells (with organohaloaluminate electrolyte); inset: XRD spectra for (1) as-fabricated tin, (2) magnesiated tin (or Mg_2_Sn—peak positions marked with arrows) and (3) de-magnesiated Mg_2_Sn. Adapted with permission from [162]. Copyright Royal Society of Chemistry, 2015. Adapted with permission from [26]. Copyright Informa UK Limited, 2017.

**Figure 9 nanomaterials-11-00810-f009:**
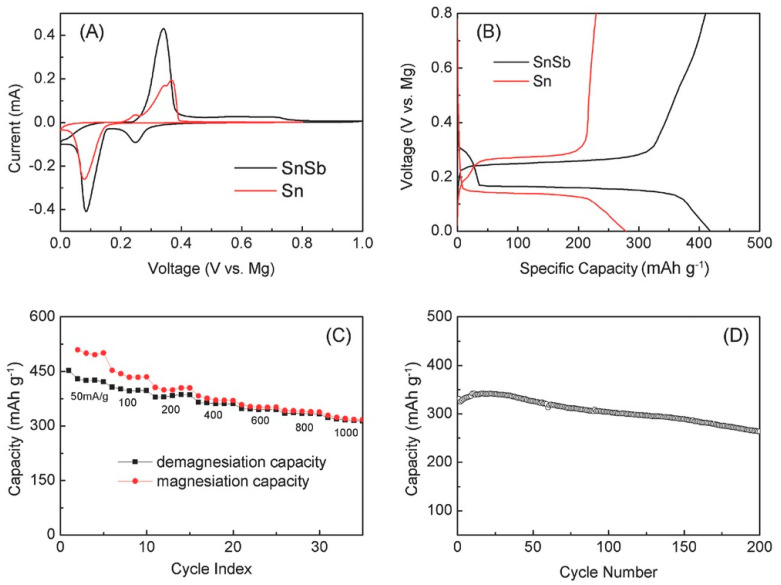
Comparison of electrochemical behaviour of tin–antimony alloy and tin towards magnesiation/de-magnesiation: (**A**) CV carried out at 0.05 mV s^−1^; (**B**) Charge–discharge profiles at 50 mA g^−1^; (**C**) Specific capacity at different current densities when using the tin–antimony alloy; (**D**) Cycling stability of the tin–antimony alloy at 500 mA g^−1^. Adapted with permission from [167]. Copyright American Chemical Society, 2015.

**Figure 10 nanomaterials-11-00810-f010:**
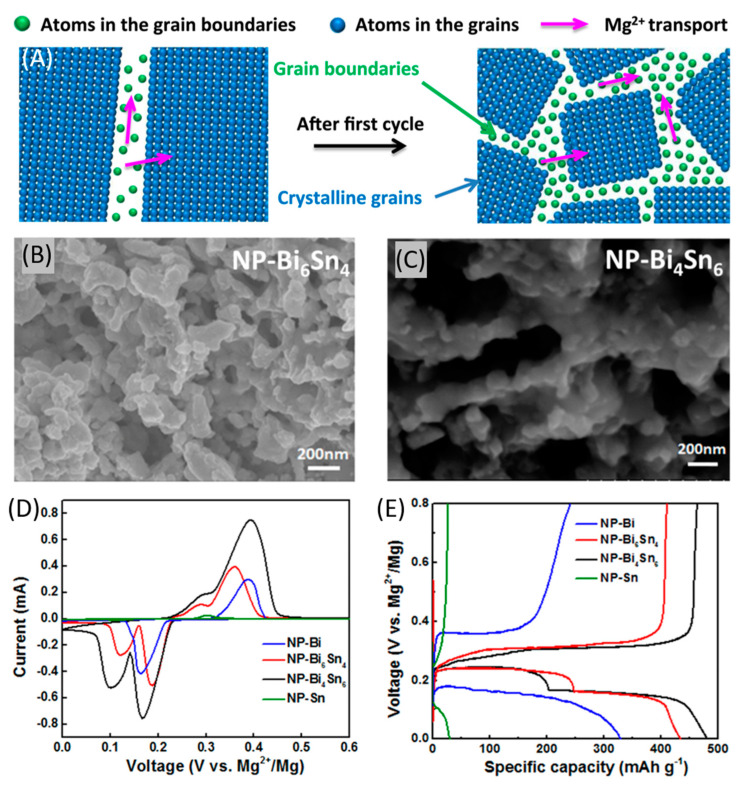
(**A**) The increased grain boundaries on the atomic scale after the first cycle and the enhanced magnesium ions transport; SEM images of (**B**) NP-Bi_6_Sn_4_ and (**C**) NP-Bi_4_Sn_6_; (**D**) CV curves for NP-bismuth and alloy electrodes at a scan rate of 0.05 mV s^−1^ and for NP-tin at 0.01 mV s^−1^ for the first cycle; (**E**) Discharge/charge profiles for NP-bismuth and alloy electrodes acquired at 50 mA g^−1^ and for NP-tin at 20 mA g^−1^. Adapted with permission from [170]. Copyright Elsevier B.V., 2018.

**Figure 11 nanomaterials-11-00810-f011:**
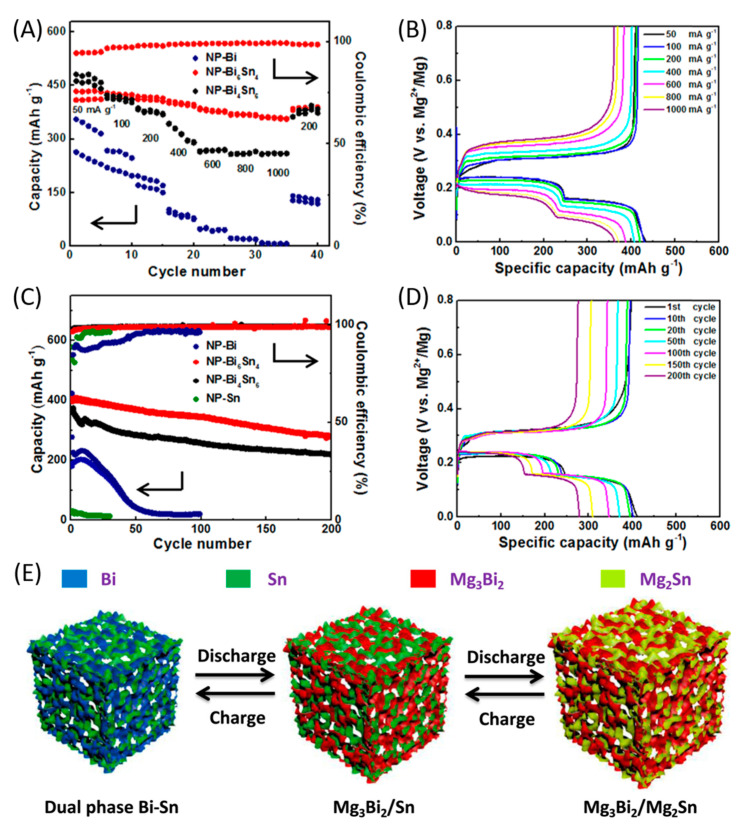
(**A**) Rate performance for NP-bismuth and alloy electrodes; (**B**) Discharge/charge profiles for NP-Bi_6_Sn_4_ electrode at different current densities; (**C**) Cycling stability of NP-bismuth and alloy electrode at 200 mA g^–1^ and of NP-tin at 20 mA g^–1^; (**D**) Discharge/charge profiles of NP-Bi_6_Sn_4_ electrode for different cycles at 200 mA g^–1^; (**E**) Schematic illustration of the electrochemical reaction mechanisms of alloy electrodes. Adapted with permission from [170]. Copyright Elsevier B.V., 2018.

**Figure 12 nanomaterials-11-00810-f012:**
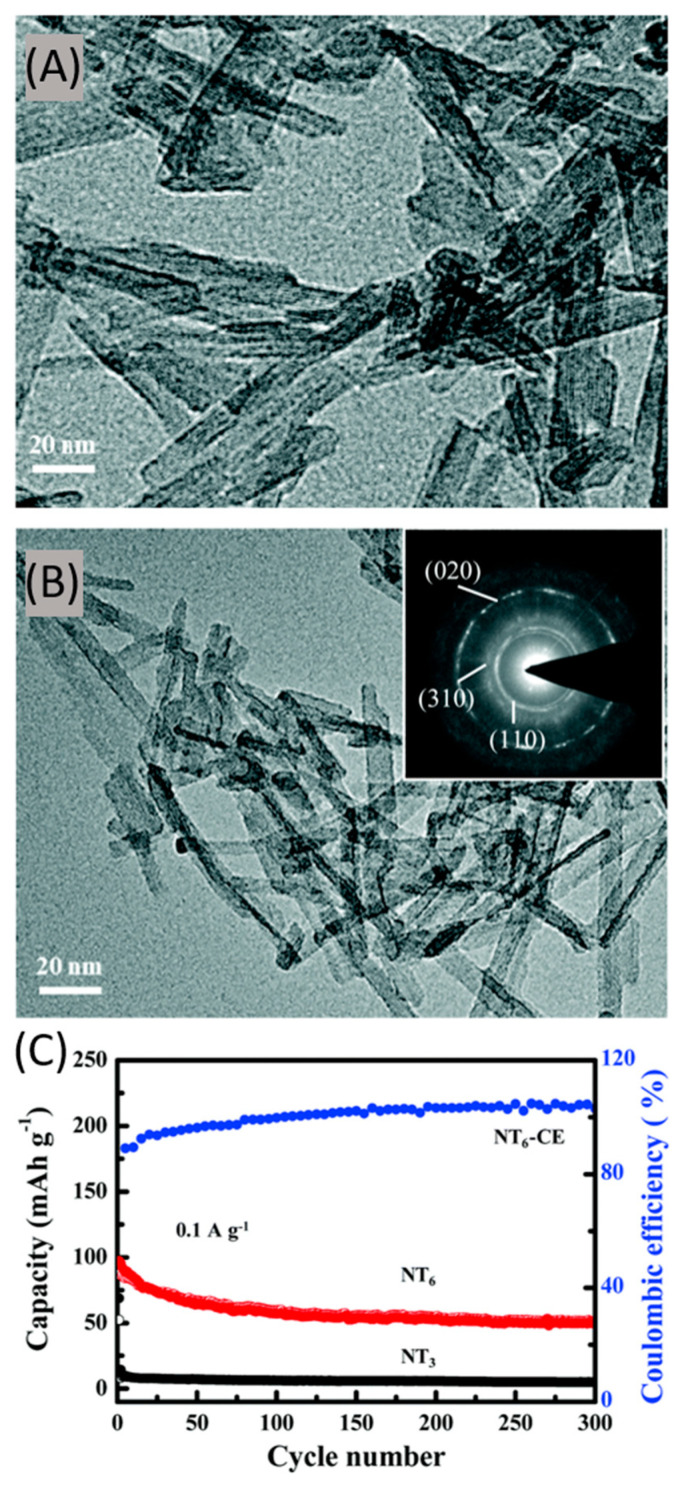
Transmission electron microscopy (TEM) images of (**A**) Na_2_Ti_3_O_7_ NWs and (**B**) Na_2_Ti_6_O_13_ NWs; (**C**) Cycling performance of Na_2_Ti_3_O_7_ and Na_2_Ti_6_O_13_ NWs at 0.1 A g^−1^. Adapted with permission from [177]. Copyright American Chemical Society, 2020.

**Figure 13 nanomaterials-11-00810-f013:**
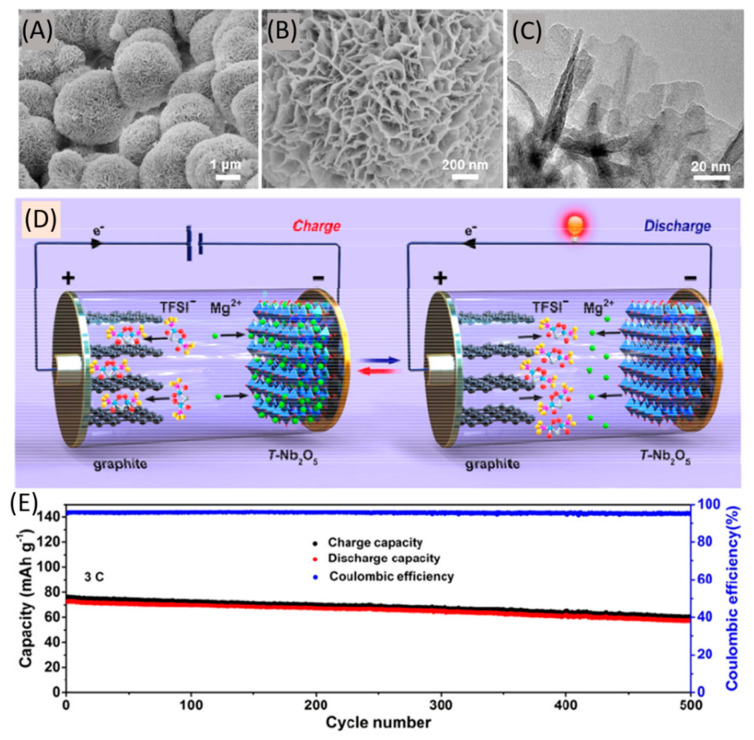
(**A**) Low- and (**B**) high-magnification SEM images and (**C**) TEM image of the Ti–Nb_2_O_5_ NFs; (**D**) Operating principle of the magnesium-ion dual-ion battery and (**E**) its long-term cycling performance at 3C with Ti–Nb_2_O_5_ NFs as anode. Adapted with permission from [177]. Copyright American Chemical Society, 2020.

**Figure 14 nanomaterials-11-00810-f014:**
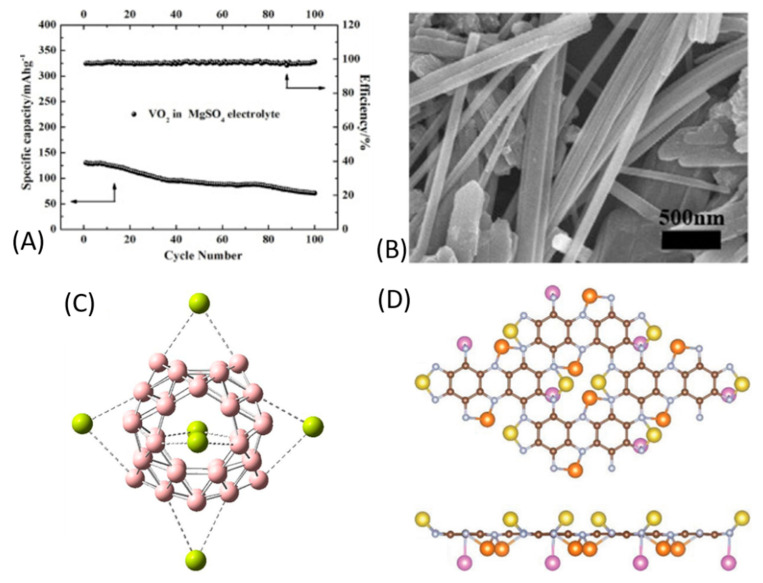
(**A**) Cycling properties of VO_2_ NWs in MgSO_4_ electrolyte; (**B**) TEM micrograph of the anode before cycling; (**C**) Relaxed geometries of fully Mg decorated of bare and halide encapsulated B_40_ (B in pink, Mg in green); (**D**) Top and side views of the most stable configuration for five Mg atoms adsorption on a C_2_N monolayer (yellow, orange and pink balls denote the Mg atoms located in the top, middle and bottom layers from the side view). Adapted with permission from [183,184,185]. Copyright Elsevier B.V. (2019,2021) and Royal Society of Chemistry (2019).

**Table 1 nanomaterials-11-00810-t001:** Main battery technologies, along with typical electrochemical performance, application and constrains. Adapted with permission from [10]. Copyright Elsevier B.V., 2014.

Battery Type	Specific Energy—Gravimetric (Wh kg^−1^)	Cycle Life (Lifetime)	Advantages	Technical and Cost Barriers
Lead acid [58]	30–50	500–1000	Low cost, mature and readily available, reliable and easily replaced, suitable for power quality, UPS and spinning reserve applications.	Short cycling capability, low power and energy density, slow charge. Low weight-to-energy ratio, thermal management requirement, environmental hazards, but fully recyclable.
Ni-Cd sealed [59]	30–45	500–800	Relatively high energy density, relatively low cycling capability, high mechanical resistance, low maintenance requirement, suitable for power tools, emergency lighting, generator starting, telecoms and portable devices.	High cost, environmental hazards, memory effect.
Ni-MH [59]	40–80	600–1200	Hybrid electric vehicles, portable electronic devices.	High self-discharge rate, low-temperature performance of the metal-hydride anode.
Na-S [58]	150–240	2500	Relatively high power and energy density, efficient, economical for power quality and peak shaving purposes.	High operating temperature (≈ 300–350 °C). Heat source requirement, high cost.
NaNiCl ZEBRA [59]	85–140	≈ 2500	Ability to withstand limited overcharge and discharge, relatively high electrochemical cell voltage (2.58 V), suitable for load-levelling applications	High operating temperature (≈ 270–350 °C), limited energy density. Lower power and energy density compared to NaS.
Vanadium redox flow battery [58]	10–30	12,000	Energy and power independent, long life cycle, low self-discharge rates. Useful for large-scale applications.	High cost, complex standardization, low energy and power density, toxic remains.
Lithium ion [10,59,60]	100–300	> 5000	Relatively high power and density, almost 100% efficient, higher cycling capacity, fast response to charge and discharge operations. Useful for laptop computers, mobile devices, hybrid electric vehicles.	Reduced first-cycle capacity loss and volumetric expansion of intermetallic electrodes. High cost, degrades at high temperatures.

**Table 2 nanomaterials-11-00810-t002:** Average abundancy in the Earth’s crust of the most commonly used elements in the batteries field. Adapted with permission from [106]. Copyright John Wiley & Sons, Inc., 1983.

Element	Average Abundancy (ppm)	Element	Average Abundancy (ppm)
Aluminium	84,249	Sulphur	404
Iron	52,157	Chromium	320
Magnesium	28,104	Zinc	72
Sodium	22,774	Copper	27
Titanium	4136	Cobalt	26.6
Manganese	774	Nickel	26.6
Phosphorus	567	Lanthanum	20
Barium	456	Lithium	16

**Table 3 nanomaterials-11-00810-t003:** Performances of Mg cells assembled with Bi and Bi_0.88_Sb_0.12_ anodes.

	Maximum Specific Capacity (mAh g^−1^)	Specific Capacity at the 100th Cycle (mAh g^−1^)
Bi	257	222
Bi_0.88_Sb_0.12_	298	215

**Table 4 nanomaterials-11-00810-t004:** Performances of Mg cells assembled with NP-Bi_4_Sn_6_ and NP-Bi_6_Sn_4_ anodes, with a focus on the contribution of each metal.

	Contribution of Bi (mAh g^−1^)	Contribution of Sn (mAh g^−1^)
NP-Bi_4_Sn_6_	202	280
NP-Bi_6_Sn_4_	246	188

**Table 5 nanomaterials-11-00810-t005:** Performances of Mg cells assembled with NP-Bi_6_Sn_4_ anode, with a focus on the contribution of each metal at different stages of the charge/discharge experiments.

Contribution of Bi to Initial Specific Capacity	Contribution of Bi to Specific Capacity at the 200th Cycle	Percentage of Capacity Fading of Bi	Contribution of Sn to Initial Specific Capacity	Contribution of Sn to Specific Capacity at the 200th Cycle	Percentage of Capacity Fading of Sn
248 mAh g^−1^	154 mAh g^−1^	37.9%	164 mAh g^−1^	126 mAh g^−1^	22.7%

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
