# Peer review of "An Overview on Anodes for Magnesium Batteries: Challenges towards a Promising Storage Solution for Renewables"

_nanomaterials, 2021, doi:10.3390/nano11030810_

Round 1

Reviewer 1 Report

The manuscript " An overview on anodes for magnesium batteries: challenges towards a promising storage solution for renewables " is very interesting and is well written. The abstract gives a concise summary of the manuscript. The results are also adequate and well analysed/evaluated. The conclusions highlighted and summarised the contents of the manuscript. The manuscript will fit really well within the scope of the magazine, therefore, I will recommend its acceptance as it is.

Author Response

We thank the Referee for handling our manuscript, for the constructive comments and for recommending the publication of this minireview as it is.

Reviewer 2 Report

The manuscript “An overview on anodes for magnesium batteries: challenges to-wards a promising storage solution for renewables” reviews the progress in the development of anode materials for Mg batteries. The introduction part presents a wide picture of the demands for rechargeable batteries pointing to the advantages of Mg-ones. The available types of Mg anodes have been examined and the advantages and disadvantages of every one of them have been outlined.  From my point of view the manuscript deserves publishing after minor revision concerning mainly the conclusion part, where reader expect to see generalized summary and some suggestion about which of the of the materials regarded in this review the Authors consider as perspective. Some suggestions about other possible solutions will be welcome too.

Another point is that the Journal is “Nanomaterials”. I think that the Authors should emphasize to the nanodimensionality of the considered materials and to try to outline and underline  their advantages.

The third point concerns the use of roman numerals for cited literature. At first glance it was charming, but with the increse of the numbers the work with the literature became perplexing and difficult.  Please, change it.

Author Response

REPORT FROM REFEREE 2

AUTHORS’ RESPONSE

The manuscript “An overview on anodes for magnesium batteries: challenges to-wards a promising storage solution for renewables” reviews the progress in the development of anode materials for Mg batteries. The introduction part presents a wide picture of the demands for rechargeable batteries pointing to the advantages of Mg-ones. The available types of Mg anodes have been examined and the advantages and disadvantages of every one of them have been outlined.  From my point of view the manuscript deserves publishing after minor revision concerning mainly the conclusion part, where reader expect to see generalized summary and some suggestion about which of the of the materials regarded in this review the Authors consider as perspective. Some suggestions about other possible solutions will be welcome too.

We thank the Referee for handling our manuscript, for the constructive comments and for recommending its publication after some revisions she/he proposed.

The manuscript has been revised according to the Reviewer’s remarks; all changes made are highlighted in the submitted file.

As regards the Conclusions section, a brief comparison between the anodes described in the review is already offered, highlighting pros/cons of Mg metal, Sn/Bi-based alloys and other materials. In addition, following the Referee’s suggestion, we have added a paragraph where we propose a couple of material classes that could deserve attention in the forthcoming years for designing new anodes for MIBs.

Another point is that the Journal is “Nanomaterials”. I think that the Authors should emphasize to the nanodimensionality of the considered materials and to try to outline and underline their advantages.

Following the Referee’s suggestion, at the end of Section 1 we have added a sentence on the role of nanodimensionality on the electrochemical performance of MIBs. Indeed, our minireview specifically discusses relevant case studies where nanomaterials act as leading cell components. A short list of the mentioned nanomaterials is reported below:

·         Bismuth nanotubes, page 10.

·         Bismuth/carbon nanotubes, page 12.

·         Nanosized tin particles, page 13.

·         Nanoporous structures, page 15.

·         Tin nanocrystals, page 19.

·         Sodium hexatitanate nanowires, page 19.

·         Ti-doped Nb2O5 nanoflakes, page 21.

The third point concerns the use of roman numerals for cited literature. At first glance it was charming, but with the increse of the numbers the work with the literature became perplexing and difficult.  Please, change it.

We understand Referee’s comment, we already agreed with the Editorial Office that they will take care of this issue during proofs editing, by placing Arabic numerals. Unfortunately, this happened when we submitted the first version of the manuscript without using the MDPI template.

Reviewer 3 Report

This paper shows a good review of anodes for magnesium batteries, there are some critical problems:

- What’s the related scope of this manuscript with Nanomaterials? Its better submit to other MDPI journals such as Materials or Batteries.

-Introduction is written simply, most recent research and innovation in bionanomaterials performances should be reviewed to show the gap of knowledge. The introduction should be extended with recent research papers.

- section of drawbacks and future could be increased quality of the manuscript.

- Should be provided with a comprehensive part between all of the anodes for magnesium batteries in the experimental and field-scale till now used.

There are some grammatical errors, please carefully check the whole manuscript.

Author Response

REPORT FROM REFEREE 3

AUTHORS’ RESPONSE

This paper shows a good review of anodes for magnesium batteries, there are some critical problems:

- What’s the related scope of this manuscript with Nanomaterials? Its better submit to other MDPI journals such as Materials or Batteries.

We thank the Referee for handling our manuscript, for the constructive comments and for recommending its publication after some revisions she/he proposed.

The manuscript has been revised according to the Reviewer’s remarks; all changes made are highlighted in the submitted file.

As regards the choice of the journal, this is not a regular manuscript, but a minireview submitted to the special issue entitled “Advances in Nanomaterials for Lithium-Ion/Post-Lithium-Ion Batteries and Supercapacitors”, bearing as keywords “Synthesis of novel positive/negative electrode materials for lithium and post-lithium systems”, “Metal anodes”, “Electrodes engineering/design” and “Processes for electrode preparation”. Thus, we think that this manuscript perfectly matches this editorial destination.

Moreover, at the end of Section 1 we have added a sentence on the role of nanodimensionality on the electrochemical performance of MIBs. Indeed, our minireview specifically discusses relevant case studies where nanomaterials act as leading cell components. A short list of the mentioned nanomaterials is reported below:

·         Bismuth nanotubes, page 10.

·         Bismuth/carbon nanotubes, page 12.

·         Nanosized tin particles, page 13.

·         Nanoporous structures, page 15.

·         Tin nanocrystals, page 19.

·         Sodium hexatitanate nanowires, page 19.

·         Ti-doped Nb2O5 nanoflakes, page 21.

- Introduction is written simply, most recent research and innovation in bionanomaterials performances should be reviewed to show the gap of knowledge. The introduction should be extended with recent research papers.

The Introduction of this manuscript presents the energy demand issue and proposes batteries as a solution to store the electricity produced by solar cells and wind farms. Then, we present the main classes of rechargeable batteries and introduce magnesium-based ones to fix some issues related to the lithium technology. Overall, this should clearly highlight the scope of this minireview and allow the reader to go through Section 2, where we address in detail the magnesium-based device.

Following the Referee’s suggestion, we checked that our Introduction already contains 101 references, 90% of which published in 2019-2021; thus, we can fairly say that this section is already rich and updated with recent papers.

Conversely, we did not catch the Referee’s comment on “bionanomaterials”, given the fact that it is a topic far from the battery field.

- section of drawbacks and future could be increased quality of the manuscript.

- Should be provided with a comprehensive part between all of the anodes for magnesium batteries in the experimental and field-scale till now used.

We agree with Referee’s comments, indeed the reader can make reference to:

·         Section 2, where we describe pros/cons of MIBs.

·         Section 5, where we have written a brief comparison between the anodes described in this minireview, highlighting pros/cons of Mg metal, Sn/Bi-based alloys and other materials. In addition, following the Referee’s suggestion, we have added a further paragraph where we propose a couple of material classes that could deserve attention in the forthcoming years for designing new anodes for MIBs.

·         We have specified in the Conclusions section that the presented contents refer to academic studies and the transition towards higher TRL levels has not been reached.

There are some grammatical errors, please carefully check the whole manuscript.

Following Reviewer’s comment, we have carefully re-read the manuscript, fixing a few typos. Overall, as also commented by the other two Referees (“very interesting and well written”), we think that the manuscript is now ready for acceptance in Nanomaterials.

Round 2

Reviewer 3 Report

The manuscript quailty has been greatly improved. I suggest that this manuscript can be published as is.